# OpenReview forum: "CoSMo-RL: Towards Trustworthy LMRMs via Joint Safety and Stability"
_ICLR.cc/2026/Conference — Submitted to ICLR 2026_

### Official Review · Reviewer_nHQj · 2025-10-28

**Soundness:** 3
**Presentation:** 3
**Contribution:** 3
**Rating:** 6
**Confidence:** 2

**Summary:**

The paper presents CoSMo-RL, a reinforcement learning framework aimed at training Large Multimodal Reasoning Models (LMRMs) that are both capable and safe. The framework introduces a two-stage RL pipeline:

* Stage 1 builds general reasoning and instruction-following ability.

* Stage 2 jointly optimizes safety, value alignment, and general reasoning using a multi-objective reward.

Training stability is enforced via Clipped Policy Gradient with Policy Drift (CPGD), and robustness is enhanced through multimodal jailbreak data augmentation. The resulting model, CoSMo-R1, demonstrates improved safety and reasoning capabilities compared to strong baselines such as GPT-4.1, Claude Opus 4, and Qwen2.5-VL-72B. Evaluations across safety, moral reasoning, and multimodal reasoning benchmarks show consistent gains.

**Strengths:**

* Unified optimization of safety and reasoning: CoSMo-RL integrates safety, value, and reasoning into a single RL framework, moving beyond the traditional post-hoc safety fine-tuning used in prior works.

* Stable and practical training design: The two-stage RL process and Clipped Policy Gradient with Policy Drift (CPGD) maintain stability while preventing reward hacking and mode collapse.

* Exposure-based robustness: Training directly on multimodal jailbreak data (text + image) improves resistance to real-world attacks instead of relying on synthetic evaluations only.

**Weaknesses:**

* Utility drop: Instruction-following performance (IF-Eval) falls from 86.3 % → 74.9 %, indicating a significant decrease in helpfulness as safety increases.

* No baseline + defense experiments: The study only compares model-level results; it does not test baseline reasoning models combined with defense modules.

* Over-safety behavior persists: CoSMo-R1 occasionally refuses benign prompts, even with the Helpful reward added. Over-refusal and excessive caution remain unsolved and may hurt usability in real-world deployment

**Questions:**

See the Weaknesses

---

### Official Review · Reviewer_ssSQ · 2025-10-31

**Soundness:** 1
**Presentation:** 2
**Contribution:** 1
**Rating:** 2
**Confidence:** 4

**Summary:**

This paper presents CoSMo-RL, a reinforcement learning framework for training LMRM that jointly optimizes safety and reasoning capability. The authors argue that safety should not be treated as an afterthought but co-evolved with general capabilities during training. They release CoSMo-R1, a model trained using this framework based on Qwen2.5-VL-72B.

**Strengths:**

Originality: Addresses timely safety degradation in multimodal reasoning models. Provides reasonable integration of staged RL, multiobjective rewards, and adversarial augmentation, though individual components are established techniques.
Quality: Evaluation across 10+ benchmarks and four model variants shows relatively consistent improvements competitive with GPT-4 and Claude. Ablations provide useful insights, but lack statistical rigor and reproducibility details.
Clarity: Well-structured with clear motivation and effective figures. Dense presentation sacrifices precision, with critical details deferred to appendix or omitted.
Significance:  Addresses real deployment needs and demonstrates safety-capability co-improvement is feasible. Incremental contribution with practical value if artifacts are released to enable community follow-up work.

**Weaknesses:**

1) Limited Baseline Comparisons
Absence of safety-specific baselines: The paper omits comparisons with recent safety-focused alignment methods, such as Safe RLHF-V [1] or LLaVA-RLHF [2], under the same evaluation protocol. Including these baselines would provide a clearer context for the contributions.
2）Overclaimed Contributions
The paper claims to present a "unified" framework for co-evolving safety and capability, but this framing overlooks prior work on multi-objective RLHF such as SafeRLHF-V [1], that similarly balances multiple objectives.
3) Cross-Model Results Lack Depth
Tables 6-8 show results on Qwen-7B, InternVL, and DeepSeek, but different benchmarks are used across models (e.g., HarmBench only for DeepSeek), making cross-model comparison impossible.

[1] Ji J, Chen X, Pan R, et al. Safe RLHF-V: Safe Reinforcement Learning from Multi-modal Human Feedback[J]. arXiv preprint arXiv:2503.17682, 2025.
[2] Sun Z, Shen S, Cao S, et al. Aligning large multimodal models with factually augmented rlhf[J]. arXiv preprint arXiv:2309.14525, 2023.

**Questions:**

1. Typo: "??" in Line 301 and 371
2. No explanation about "safety rate" in line 300
3. The paper states Stage 2 "jointly optimizes safety, value, and general capability" but doesn't specify. Are these trained on separate data batches or mixed? What's the sampling ratio across task types? How is catastrophic forgetting of Stage 1 capabilities prevented?

---

### Official Review · Reviewer_gTZQ · 2025-10-31

**Soundness:** 2
**Presentation:** 2
**Contribution:** 1
**Rating:** 2
**Confidence:** 3

**Summary:**

This paper aims to address the prevalent "safety tax" problem in Large Multimodal Reasoning Models (LMRMs), where enhancing model safety (e.g., through alignment) often leads to a decline in reasoning capabilities or over-refusal on benign prompts. The authors propose a hybrid Reinforcement Learning (RL) framework named COSMO-RL. Its core idea is the co-evolution of safety and general capabilities, rather than treating safety alignment as a post-hoc step.

The main components of this framework include:
1.  **Two-stage RL Training:** Stage 1 focuses on building broad general reasoning capabilities; Stage 2 then jointly optimizes for safety and value alignment while maintaining general capabilities.
2.  **CPGD Optimization Algorithm:** Utilizes the CPGD (Clipped Policy Gradient Optimization with Policy Drift) algorithm to ensure the stability of RL training.
3.  **Multi-objective Reward Function:** A unified, four-part reward function is designed, including $R_{visual-focus}$ (visual focus), $R_{helpful}$ (helpfulness), $R_{format}$ (format), and $R_{task-aware}$ (task-awareness).
4.  **Specialized Outcome Reward Models (ORMs):** To implement $R_{task-aware}$, the framework relies on three independently trained reward models: a Safety ORM, a Value ORM, and a Knowledge ORM.

The model trained through this framework, CoSMo-R1 (based on Qwen2.5-VL-72B), demonstrates stronger safety, a lower unnecessary refusal rate, and enhanced multimodal reasoning capabilities in experiments.

**Strengths:**

1.  **Importance of the Problem:** This paper addresses a very important and timely challenge in the LMRM domain. As model capabilities increase, ensuring they remain safe and controllable without sacrificing their core reasoning abilities is a critical issue. Jailbreak risks specific to multimodality also make this problem more challenging.
2.  **Comprehensive Framework Design:** The authors propose an ambitious and systematic solution. COSMO-RL is a complex, end-to-end process that integrates CoT-SFT, two-stage RL, a stable optimizer, and a sophisticated multi-objective reward mechanism, demonstrating the authors' thorough consideration in tackling this problem.
3.  **Significant Improvements on Some Metrics:** The experimental results show that CoSMo-R1's safety rate on multiple safety benchmarks (e.g., MM-SafetyBench and SIUO) far exceeds that of the baseline models. Concurrently, on general reasoning benchmarks (e.g., Olympiad and GPQA Diamond), CoSMo-R1 also demonstrates stronger performance than the baseline Qwen2.5-VL-72B.

**Weaknesses:**

1.  **Core Claim Contradicts Data; "Safety Tax" Persists:** The paper's core claim is that it "resolves" the trade-off between safety and capability, asserting that CoSMo-R1 "maintains or even enhances" capability. However, the authors state at the end of Section 4.2.3 that CoSMo-R1 achieves only 74.9% on the instruction-following benchmark IF-Eval, whereas its baseline model (Qwen2.5-VL-72B) scores 86.3%. This is a **significant performance drop of 11.4%**. The authors describe this as "no significant drop," which is a severe misinterpretation of the data. This indicates the "safety tax" has not disappeared but has **merely been shifted from "reasoning capability" to "instruction-following capability,"** which is equally detrimental for building a useful assistant.
2.  **Method is Extremely Complex and Difficult to Reproduce:** The reproduction cost of this framework is extremely high. It not only requires a complex two-stage RL pipeline and the CPGD algorithm but also **relies on three independently trained, powerful reward models** (Safety ORM, Value ORM, Knowledge ORM). These ORMs are complex models in themselves (e.g., the Value ORM is built on Qwen2.5-VL-72B). This complexity makes the framework difficult to adopt widely and is far from the "a simple path" claimed in the abstract.
3.  **Lack of Transparency and Replication Details for Core Components (ORMs):** The success of the COSMO-RL framework relies heavily on three independent "Outcome Reward Models" (Safety ORM, Value ORM, Knowledge ORM), which collectively form the $R_{task-aware}$ reward signal. However, the training and deployment details for these three critical models are severely lacking:
    * **Opaque Datasets:** The paper briefly describes the data used to train the ORMs in Appendix A.1.1, such as "80k bilingual multimodal samples" for the Value ORM and "120k multimodal knowledge questions" for the Knowledge ORM. However, these datasets are not released, and their construction processes (e.g., "expert curation" and "closed-loop process") are vague. This makes it impossible for reviewers to assess potential biases within these ORMs.
    * **Missing Training Details:** The paper mentions the base models for the ORMs (e.g., Qwen2.5-VL-7B and Qwen2.5-VL-72B) but fails to provide sufficient details on training hyperparameters, architectural modifications, or validation procedures. For example, the base model for the Knowledge ORM is not even explicitly mentioned.

**Questions:**

1.  Given that the success of the COSMO-RL framework is highly dependent on three opaque ORMs, can the authors provide more detailed ablation studies to isolate the impact of the ORM quality? For example, if simpler, more transparent reward models (e.g., keyword detection or GPT-4-based API calls) were used for the $R_{task-aware}$ signal, would the COSMO-RL two-stage training framework still be effective?

2.  Given that the framework requires training three additional large reward models (e.g., the Value ORM is based on a 72B model), can the authors detail the total computational cost required to reproduce this framework (e.g., compared to training only the baseline SFT model)?

---

### Official Review · Reviewer_GN8h · 2025-11-01

**Soundness:** 3
**Presentation:** 3
**Contribution:** 3
**Rating:** 4
**Confidence:** 5

**Summary:**

The topic of this paper is the large multimodal reasoning model. The authors train CoSMo-R1 via CosMo-RL, a mixed RL framework that trains the models under multimodal, multitask, and multiobjective signals. The experiments show the performance and the safety the proposed model.

**Strengths:**

1. The motivation is clear, and the topic is practical. Training a better and safer LMRM is significant.
2. The principles in the intro. part are interesting and can be used in other domains.
3. The experiments and analyses, e.g., case studies, are comprehensive.

**Weaknesses:**

1. The four principles in the introduction part are interesting. Could the authors provide more insights and experimental evidence?
2. Some important details are missing in the main text. I recommend that the author move the detailed calculation of the different rewards to the main text. Besides, could the authors provide more details about the Safety ORM in Table 5?
3. In Table 1, the results of DeepSeek's and Kimi's models are missing. In Table 8, the results of OpenAI's, Google's, and Anthropic's models are missing.
4. The source code and training data are missing. Will the authors release them?
5. The performance of the proposed model under the jailbreak attacks is missing, e.g., AutoDAN, ReNeLLM, FlipAttack.
6. Minor: missing discussion on [1] in the related work part. The notation table is missing. In Line 301, the number of the table is missing. In Line 370, the number of section is missing ('??').

[1] GuardReasoner-VL: Safeguarding VLMs via Reinforced Reasoning

**Questions:**

See Weaknesses.

---

### Meta-Review · Area_Chair_Poo6 · 2026-01-07

**Summary:**

The paper proposes CoSMo-RL, a hybrid reinforcement learning framework to train Large Multimodal Reasoning Models (LMRMs) that co-evolve safety and reasoning capabilities. The approach involves a two-stage RL training process and a multi-objective reward function. While reviewers acknowledged the timely motivation and comprehensive framework design, the submission was widely criticized for significant flaws. The core claim of resolving the "safety tax" is contradicted by experimental data showing a drastic drop in instruction-following capability. Additionally, the method's complexity, reliance on opaque reward models, and lack of key baseline comparisons raised serious concerns about reproducibility and contribution.

**Reviewer Concerns:**

Contradictory Claims (Reviewers gTZQ, nHQj): The paper claims to resolve the safety-capability trade-off, but results show a severe drop (11.4%) in instruction-following performance (IF-Eval), indicating the "safety tax" has merely shifted rather than disappeared.

Complexity & Reproducibility (Reviewers gTZQ, GN8h): The framework requires three independent, opaque reward models (ORMs) and a complex two-stage pipeline. The lack of open-source artifacts (code, data, ORMs) makes reproduction nearly impossible.

Missing Baselines (Reviewers ssSQ, GN8h, nHQj): Key safety-focused baselines (e.g., Safe RLHF-V, LLaVA-RLHF) and jailbreak attack evaluations (AutoDAN, etc.) are absent, making it difficult to contextualize the contribution.

Transparency Issues (Reviewer gTZQ): The training details and datasets for the critical Reward Models are undescribed, raising concerns about hidden biases or engineering hacks.

**Reviewer Scores:**

Reviewer GN8h: 4 (Borderline Reject)

Reviewer gTZQ: 2 (Reject)

Reviewer ssSQ: 2 (Reject)

Reviewer nHQj: 6 (Borderline Accept)

---

### Decision · Program_Chairs · 2026-01-26

Reject